# LADI v2: Multi-label Dataset and Classifiers for Low-Altitude Disaster Imagery

## Abstract

ML-based computer vision models are promising tools for supporting emergency management operations following natural disasters. Imagery taken from small manned and unmanned aircraft can be available soon after a disaster and provide valuable information from multiple perspectives for situational awareness and damage assessment applications. However, emergency managers often face challenges in effectively utilizing this data due to the difficulties in finding the most relevant imagery among the tens of thousands of images that may be taken after an event. Despite this promise, there is still a lack of training data for imagery of this type from multiple perspectives and for multiple hazard types. To address this, we present the LADI v2 (Low Altitude Disaster Imagery version 2) dataset, a curated set of about 10,000 disaster images captured by the Civil Air Patrol (CAP) in response to over 50 disaster events (2015-2023) from over 30 US states and territories and annotated for multi-label classification by trained CAP volunteers. We also provide two pretrained baseline classifiers and compare their performance to state-of-the-art vision-language models in multi-label classification. The data and code are released publicly to support the development of computer vision models for emergency management research and applications.

## 1 Introduction

Rapid and accurate assessment of post-disaster conditions is critical for effective disaster response and recovery operations. Low altitude aerial imagery, such as aerial photographs collected by the Civil Air Patrol (CAP) in the United States or images from small Unmanned Aerial Systems (sUAS), can provide valuable information about the extent and severity of damage caused by disasters. However, the large quantity of images collected during these missions can present a significant challenge for analysts tasked with identifying actionable information in a timely manner. To address this challenge, we introduce the Low Altitude Disaster Imagery v2 (LADI v2) dataset, a multi-label image classification dataset designed to facilitate the development of computer vision models for identifying useful post-disaster aerial images. LADI v2 builds upon existing work in image-based damage assessment (Kyrkou and Theocharides, 2020; Gupta et al., 2019; Rahnemoonfar et al., 2021; 2022; Weber et al., 2023; Liu et al., 2019) by providing a diverse, multi-hazard dataset that includes oblique and nadir aerial imagery from various locations and disaster declarations across the United States.

The technical contributions of LADI v2 are twofold. First, we present a curated dataset of post-disaster aerial imagery from multiple perspectives for multiple hazard types, with high quality, operationally-relevant labels annotated by trained CAP volunteers for multi-label classification. Second, we provide two pretrained baseline reference classifiers. We also demonstrate the utility of the dataset as a benchmark, and compare the performance of the baseline classifier to state-of-the art vision-language models (VLM) on open-vocabulary classification as a benchmark. We outperform the open source VLM on nearly all classes and are competitive with the commercial VLM in test set, and broadly outperform both VLMs in the validation set. This demonstrates the continued need for open, domain-specific training data for specialized applications such as disaster response.

LADI v2 also offers unique characteristics that can contribute to the advancement of machine learning research. The dataset features a realistic distribution shift between the training and test sets, representing annual variation in disaster incident types, as well as changes due to new operational procedures and technologies. This characteristic makes LADI v2 a valuable benchmark for evaluating domain adaptation techniques in the context of disaster response.

The dataset, classifiers, and associated documentation are made openly available on GitHub (Anonymized Authors, 2024b) and Hugging Face (Anonymized Authors, 2024c;a), enabling researchers and practitioners to build upon this work and adapt the models to their specific needs. Through this contribution, we aim to streamline the process of identifying useful post-disaster aerial images, ultimately supporting more efficient and effective disaster response efforts while providing a valuable resource for the broader machine learning community.

The paper is structured as follows: Section 2 covers related work in disaster imagery datasets and necessary background on vision-language models; Section 3 provides the details of the dataset; Section 4 discusses the pretrained baseline classifiers; and Section 5 concludes with a summary and comments on limitations and future directions.

## 2 Related Work

### 2.1 Natural Disaster Imagery Datasets

There is existing work on using imagery and computer vision to facilitate post disaster damage assessment (Kyrkou and Theocharides, 2020; Gupta et al., 2019; Rahnemoonfar et al., 2021; 2022; Weber et al., 2023; Manzini et al., 2024; Liu et al., 2019). We highlight a few relevant examples and comment on the gap that LADI v2 addresses. The xBD dataset (Gupta et al., 2019) provides 23,000 labeled nadir-perspective satellite images annotated with bounding boxes for building damage for multiple locations across the globe, covering multiple hazard types. FloodNet (Rahnemoonfar et al., 2021) and RescueNet (Rahnemoonfar et al., 2022) each provide segmentation masks, classification labels, and visual question-answering captions for high resolution, low altitude aerial imagery from UAVs, but are limited to single incidents each: Hurricanes Harvey and Michael, respectively. Incidents1M (Weber et al., 2023) provides a large multi-label dataset classifying 43 incident types in 49 outdoor locations from nearly 1 million images scraped from the web. These images feature a variety of perspectives and locations, but are primarily ground-based, and only identify the type and location of the incident. CRASAR-U-DROIDs (Manzini et al., 2024) provides annotated building damage polygons for orthomosaic imagery from small Unmanned Aerial Systems (sUAS). Compared to the existing literature, there is still a lack of training data to support both oblique and nadir low-altitude aerial imagery—which is increasingly common in disaster response applications with UAVs and small manned aircraft—across multiple geographies, event types, and damage and infrastructure categories.

Version 1 of the Low Altitude Disaster Imagery dataset (LADI v1) began to address these gaps by providing annotations for low altitude, multi-perspective imagery from multi-hazard and multi-geographic events (Liu et al., 2019). It was included in a number of NIST TRECVID challenges (Awad et al., 2021a;b; 2023). However, these labels were created by untrained crowdsourced workers, and the term "damage" was not clearly defined. Instead, LADI v2 was labeled by a team of volunteer annotators from the Civil Air Patrol who have been trained in the FEMA damage assessment process. Damage labels follow FEMA's criteria for preliminary damage assessments (PDAs), which articulate five levels of damage: unaffected, affected, minor damage, major damage, and destroyed. Furthermore, the label set and annotator training materials were developed in conjunction with FEMA's Response Geospatial Office to ensure compatibility with their procedures. For these reasons, LADI v2 offers a different label set than LADI v1. Nevertheless, we still believe LADI v1 offers some value, since it features a larger number of annotated images, and thus may serve as a suitable pretraining task for classifiers trained on LADI v2.

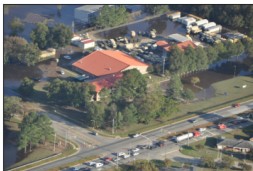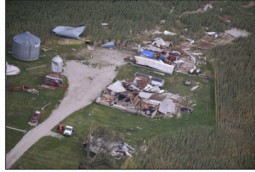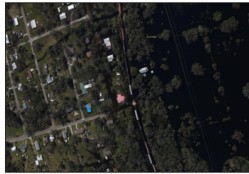

Figure 1: A sample of images from our training set. LADI v2 contains images with both positive and negative examples of damage from a range of altitudes, perspectives, geographies, and lighting conditions

## 2.2 Vision-Language Models

Recent advances in vision-language models represent extremely promising developments toward generalizable solutions for computer vision problems (Gan et al., 2022). Vision-language models typically include an image encoder and text encoder for each respective data modality. These models are trained on tasks to promote alignment between the image encoding and text encoding; such tasks include contrastive image-text learning, popularized by (Radford et al., 2021), as well as input reconstruction such as masked language modeling (Devlin et al., 2018) and masked image modeling (Tan and Bansal, 2019). These vision-language models can be used to address various computer vision tasks, including open vocabulary classification, object detection, and segmentation (Gan et al., 2022). We discuss a few vision language models that are relevant to this paper, particularly LLaVA-NeXT (Liu et al., 2024), and GPT-4o (OpenAI, 2024).

LLaVA (Large Language and Vision Assistant) (Liu et al., 2023b) and its refinements LLaVA 1.5 (Liu et al., 2023a) and LLaVA-NeXT (Liu et al., 2024) build upon CLIP by incorporating a large language model decoder, such as Vicuna (Chiang et al., 2023) or Mistral (Jiang et al., 2023), to allow it to be trained additional tasks, such as visual question answering. Commercial offerings such as GPT-4o (OpenAI, 2024) offer similar multimodal capabilities. We benchmark our models on LADI v2 against zero-shot classification using LLaVA-NeXT and GPT-4o in Section 4.4.

# 3 LADI v2 Dataset

## 3.1 Data collection and annotation

LADI v2 images are sourced from the FEMA (Federal Emergency Management Agency) Civil Air Patrol Image Uploader Repository, publicly hosted in an Amazon Web Services s3 bucket (FEMA, 2024a), which contains aerial photographs collected in support of federally declared disasters from 2009 onward, as well as CAP training missions. Figure 1 shows a sample of images in the dataset. Image metadata includes timestamp and location information. To ensure LADI v2 contains only images collected during disasters, we compared image metadata against disaster declaration data from the OpenFEMA Disaster Declarations API (FEMA, 2024b) to identify images taken within 14 days of the start of a declared federal disaster and within an affected county's boundaries.

We consulted emergency management professionals at the FEMA Response Geospatial Office to develop the set of labels and labeling instructions. Labels were chosen to help emergency managers identify the most relevant images when conducting initial damage assessments, which can support disaster declarations and assistance grants. The label set is provided as the "v2" set in Table 2a. The various damage levels for the buildings—affected, minor, major, and destroyed, listed in increasing order of severity—are determined based on the FEMA preliminary damage assessment criteria (FEMA, 2021).

Images were annotated by a team of 46 Civil Air Patrol volunteers who had been previously trained in the FEMA preliminary damage assessment process (FEMA, 2021). Each image was shown to three volunteers, and labels were assigned by majority vote when there was disagreement between the annotators.

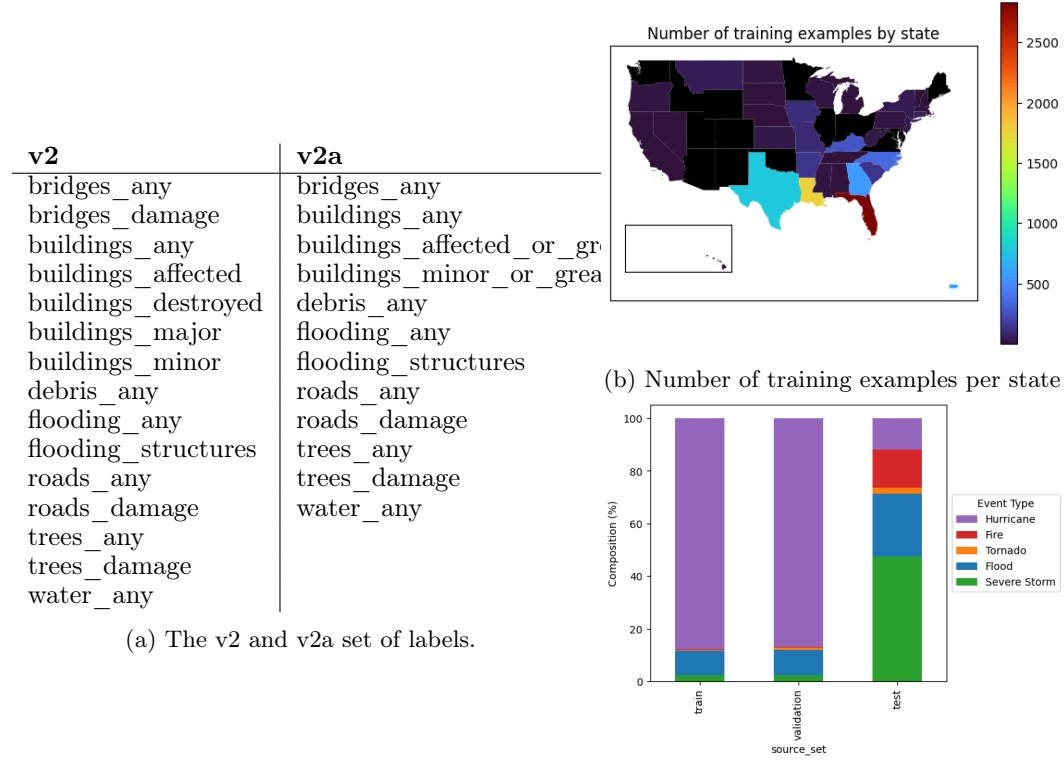

| v2 | v2a |
|---|---|
| bridges_any | bridges_any |
| bridges_damage | buildings_any |
| buildings_any | buildings_affected_or_gr... |
| buildings_affected | buildings_minor_or_grea... |
| buildings_destroyed | debris_any |
| buildings_major | flooding_any |
| buildings_minor | flooding_structures |
| debris_any | roads_any |
| flooding_any | roads_damage |
| flooding_structures | trees_any |
| roads_any | trees_damage |
| roads_damage | water_any |
| trees_any | |
| trees_damage | |
| water_any | |

(a) The v2 and v2a set of labels.

(b) Number of training examples per state

(c) Distribution of events in the dataset splits

Figure 2: Details of the dataset: the label sets, number of training examples by state, and event type distribution of the various splits.

After initial experiments with training classifiers on the "v2" label set, we found that there were very few positive examples of "bridges_damage". In addition, we found that performance was poor when distinguishing between various levels of building damage ("buildings_affected", "buildings_minor", "buildings_major", and "buildings_destroyed"). FEMA staff advised that it was less important to classify building damage categories than to determine whether buildings that had sustained any level of damage were present in image. In light of this, we removed the "bridges_damage" class and combined the building damage categories into "buildings_affected_or_greater" and "buildings_minor_or_greater". This revised set of labels is called the "v2a" label set, as shown in Table 2a, and is what we report our results on. The "v2a" label set contains 12 labels, categorized into 5 non-damage-related ("bridges_any", "buildings_any", "roads_any", "trees_any", and "water_any") and 7 damage-related ("buildings_affected_or_greater", "buildings_minor_or_greater", "debris_any", "flooding_any", "flooding_structures", "roads_damage", and "trees_damage").

## 3.2 DATASET STATISTICS AND CHARACTERISTICS

Our dataset consists of 9,963 images, split into 8,030 train examples, 892 validation examples, and 1,041 test examples. The training and validation examples are drawn from disaster declarations between 2015-2022, and the test examples are drawn from disaster declarations in 2023. In total, the dataset draws from over 100 disaster declarations from more than 30 US states and territories, see Figure 2. Since disaster declarations can span multiple states, we estimate the number of distinct disaster events by clustering the declarations based on incident date, and we estimate that at least 50 distinct disaster events are in the dataset.

*Disaster type distribution.* Since the training and validation examples are drawn from the same set of disaster incidents, the distributions of hazard types for those two splits are quite similar, whereas the test set has comparatively more images from severe storms, floods, and

fires, and many fewer images from hurricanes. The hazard type distributions of the splits are visualized in Figure 2c.

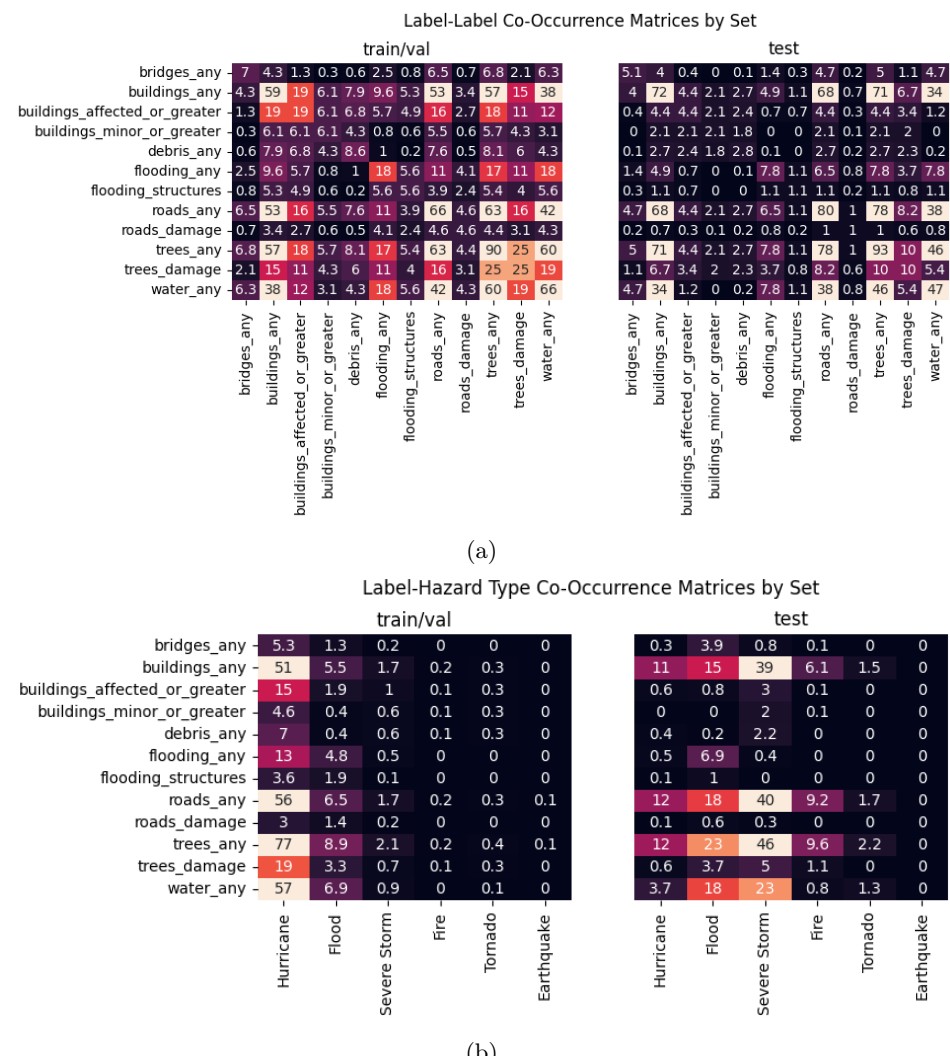

(a)

(b)

Figure 3: Co-occurrence matrices for data splits. Numbers indicate percentage of images within the given split that have the given combination of labels. Lighter colors indicate higher percentages. Note that training and validation sets are combined in these figures for brevity due to their similar distributions.

*Label distribution and co-occurrence.* Figures 3a and 3b give the label-label co-occurrence matrices and label-hazard type co-occurrence matrices for the training, validation, and test sets. As the training and validation sets are randomly drawn from the same disaster incidents, they do not have substantially different statistical distributions and are combined in these figures for the sake of brevity. We observe a substantial difference between the training/validation co-occurrence matrices and the test co-occurrence matrices. In the label-hazard type co-occurrence matrix, this is easily explained by the fact that different incident hazard types were more common in the 2023 test set. The label-label matrices require slightly more in-depth explanation.

The label-label matrices indicate that images showing damage occur less frequently in the test set. We believe this is due to a combination of factors. First, there is natural variation in the intensity and distribution of damage for different incidents. Second, CAP has recently expanded their use of the WaldoAir camera system (Patrol, 2023). This system takes images at regular time intervals in a grid pattern (Patrol, 2023), whereas images taken from handheld

cameras tend to focus on pre-selected targets or areas that the photographer selects, which often are areas with prominent damage. As a consequence, a much lower percentage of the WaldoAir images contain damage. The train and validation sets consist of about 50% Waldo images, while the test set consists of 65% Waldo images.

*Distribution shift.* The distribution shifts between the train/validation and test sets represent challenges in disaster applying for machine learning. The distribution of hazard types, severity, and locations change year to year based on cyclical weather patterns (Kovats et al., 2003) and climate change (Holland and Bruyère, 2014). Furthermore, changes in operational procedures and technology, such as the increased adoption of the WaldoAir system, can lead to distribution variation even among disasters resulting from the same type of hazard. This underscores one of the key domain-specific challenges of applying machine learning solutions for disaster response applications. To our knowledge, no other benchmark disaster imagery dataset explicitly addresses the shift in data distribution from year to year.

### 3.3 COMPARISON TO EXISTING DATASETS

Compared to LADI v1 (Liu et al., 2019), v2 offers higher-quality labels from trained annotators, and a different label set. While both versions of the dataset draw from the same repository of public domain operational FEMA CAP images (FEMA, 2024a), only about 2.4% of images from v2 also appear in v1.

The label set for LADI v2 includes building damage on the FEMA PDA scale, which is compatible with the damage scale used by xBD (Gupta et al., 2019), RescueNet (Rahnemoonfar et al., 2022), and CRASAR-U-DROIDs (Manzini et al., 2024). Compared to those datasets, LADI v2 includes more distinct events (LADI v2: 50+, xBD: 19, RescueNet: 1, CRASAR-U-DROIDs: 10), total pixels: (LADI v2: 345.32e9, xBD: 23.14e9, RescueNet 53.99e9, CRASAR-U-DROIDs: 67.13e9), and among aerial datasets, area covered (LADI v2: 161.4 km$^2$, RescueNet: 3.6 km$^2$, CRASAR-U-DROIDs: 67.98 km$^2$)[1]. LADI is also unique in its inclusion of both oblique and nadir perspective imagery. However, LADI v2's labels only support image classification, whereas the other mentioned datasets provide segmentation polygons for building damage. Thus, LADI v2 serves to complement the capabilities of those existing datasets, and would be an ideal candidate to include in a multi-task training framework, especially due to the alignment in building damage labels.

## 4 PRETRAINED CLASSIFIERS

### 4.1 ARCHITECTURE AND TRAINING DETAILS

To support research and deployment applications, we provide two pretrained reference classifiers, LADI-v2-classifier-small-reference and LADI-v2-classifier-large-reference,[2] hereby referred to as the "small" and "large" classifiers for brevity. The small classifier is based on the Big-Transfer (BiT-50) architecture (Beyer et al., 2022), pretrained on ImageNet-1k (Deng et al., 2009), while the large classifier is based on Swin v2 Large (Liu et al., 2021), pretrained on ImageNet-21k (Ridnik et al., 2021) and finetuned on ImageNet-1k (Deng et al., 2009). Standard random augmentations of resizing, cropping, horizontal flipping, affine transformations, and color jitter were applied to the training images.

These architectures were the best performers among a number of available models. We first finetuned twenty five pretrained classifiers, available on Hugging Face, on LADI v2 using default settings. We chose the top two performing candidates and performed additional optimization through hyperparameter tuning and pretraining on LADI v1 (Liu et al., 2019). Hyperparameter tuning was done via random search on optimizer type, learning rate, and learning rate scheduler. We considered the AdamW (Loshchilov and Hutter, 2019) and

---

[1]xBD is a satellite imagery dataset, which covers much wider area at lower resolution. It covers over 45000 km$^2$

[2]We provide four classifiers in the repository. The "reference" versions, discussed in his paper, are trained only on the train set. The "main" versions are trained on all splits and intended for deployment and downstream applications.

| Name | Architecture | LR Scheduler | Optimizer | Initial LR | $\gamma$ | Epochs |
|---|---|---|---|---|---|---|
| Large | Swin v2 | Exponential | AdamW | $5 * 10^{-5}$ | 0.9 | 50 |
| Small | BiT-50 | Exponential | AdamW | $1 * 10^{-4}$ | 0.9 | 50 |

Table 1: Selected hyperparameters for our reference small and large models.

Adafactor (Shazeer and Stern, 2018) optimizers, initial learning rates of $2 \times 10^{-4}$, $1 \times 10^{-4}$, and $5 \times 10^{-5}$, and LR schedulers with exponential LR decay ($\gamma$: 0.5 or 0.9) or with reduce LR on plateau ($\gamma$: 0.5 or 0.9, patience: 5 or 10 epochs). Results for the top two architectures are shown in ablation in Table 2; we evaluate using mean Average Precision (mAP).

The final configurations for LADI-v2-classifier-small and LADI-v2-classifier-large are indicated in Table 2 with † and ‡ respectively. Hyperparameter tuning provides significant benefit for both architectures. Pretraining on LADI v1 provides modest benefits in the test mAP for the model based on Swin v2, but degrades performance for the model based on BiT-50. This is consistent with the observations of Beyer et al. (2022), who find that pretraining BiT-50 on larger datasets does not improve, but instead degrades fine-tuning performance.

For the sake of brevity, all subsequent results are presented only for the "large" model.

| Architecture | Tuning | v1 Pretrain | Validation mAP | Test mAP |
|---|---|---|---|---|
| BiT-50 | No | No | 89.6 | 89.0 |
| Swin v2 | No | No | 88.7 | 86.0 |
| †BiT-50 | Yes | No | 93.3 | 91.1 |
| Swin v2 | Yes | No | **93.8** | 92.3 |
| BiT-50 | Yes | Yes | 87.9 | 85.9 |
| ‡Swin v2 | Yes | Yes | **93.8** | **92.6** |

Table 2: Classifier performance ablation with hyperparameter tuning and LADI v1 pretraining. The small model configuration is indicated with † and the large model with ‡.

## 4.2 DIRECT COMPARISON AGAINST LADI v1

Although the label sets of LADI v1 and v2 differ, we developed a mapping that condenses both into a common set of classes to facilitate a direct comparison. This mapping enables us to assess the marginal benefit of using trained annotators over crowdsourced workers[3]. The mapping is presented in the appendix (Table 5), where the condensed classes are 'building', 'flooding', 'road', 'damage', and 'debris'. We evaluated both models on LADI v2's validation and test splits, with results displayed in Table 3. In all metrics, the model trained on LADI v2 outperformed, highlighting the advantages of higher-quality labels provided by trained CAP annotators.

| split | version | mean_prec | mean_rec | mean_f1 | mean_ap |
|---|---|---|---|---|---|
| val | v1 | 0.799 | 0.808 | 0.800 | 0.869 |
| val | v2 | **0.877** | **0.895** | **0.886** | **0.946** |
| test | v1 | 0.850 | 0.768 | 0.800 | 0.877 |
| test | v2 | **0.871** | **0.850** | **0.859** | **0.917** |

Table 3: Performance of the large classifier on the v2 test and validation sets when trained on the LADI v1 train set versus the LADI v2 train set, as measured by mean precision, recall, F1 score, and mean average precision. The top scores for each split are bolded.

---

[3]This analysis does not fully account for other enhancements from v1 to v2, such as the more standardized and detailed damage labels.

### 4.3 Performance Analysis

*Incident hazard type analysis.* The performance between the test and validation sets is comparable for most hazard types, except for fire (see Figure 4a). The difference in performance for fire incidents is likely due to the relative lack of fire data in the training and validation sets compared to the test set.

*Geographic analysis.* The classifier performs robustly across various geographies. Figure 4b shows the mAP of the classifier for states in the test set, which are those states in which CAP collected images for disasters in 2023. While only 10 states are represented in the test set, the classifier achieved an mAP between 80 and 96 for all of them. The state of Hawaii, with the lowest mAP at 80, had only one CAP mission in the test set, the August 2023 Hawaii wildfires. Relatively low performance is likely due to the relative lack of fire events in the training data, as well as potential differences due to geography. We thus caution practitioners and researchers against using models trained on LADI v2 for applications not well represented in the training set and recommend supplemental data collection and training.

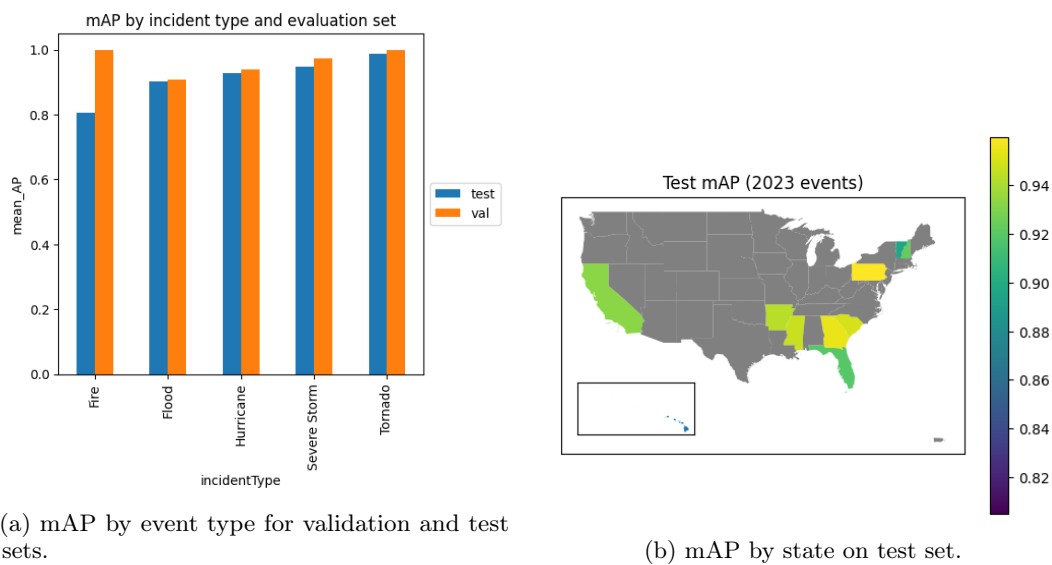

(a) mAP by event type for validation and test sets.

(b) mAP by state on test set.

Figure 4: Characterization of classifier performance by event type and location.

### 4.4 Benchmarking open-vocabulary classification (LLaVA and GPT-4o)

LADI v2 also has potential to be an effective benchmark for supervised and zero-shot classification of post-disaster imagery. In particular, the validation and test sets of LADI v2 offer the ability to test model performance against a realistic distribution shift observed in practice. We demonstrate this by comparing our model's performance to recent open-vocabulary classifiers.

We first evaluate our model compared to the recently released open 7.5 billion parameter LLaVA-NeXT model (Liu et al., 2024) on a zero-shot classification task. For each class, the LLaVA model saw an image in the test or validation set, followed by a prompt such as, "Does this image contain `class_name`? Answer with 'yes' or 'no.'" For the classes involving FEMA preliminary damage assessment categories, we included a summary of the damage category criteria based on the FEMA Preliminary Damage Assessment Pocket Guide (FEMA, 2021) in the prompt. The model outputs were converted to binary labels and used to compute the $F_1$ score and shown in Table 4. In the test set, our method outperformed LLaVA-NeXT on all labels, including all damage-related classes, except in three categories: "bridges_any", "roads_any", and "trees_any", in which our model comes within 3% of LLaVA-NeXT. In the validation set, our model outperforms LLaVA-NeXT in all categories.

We also compare our model to a commercial multimodal model, GPT-4o, using the same prompt format as above. The results are shown in Table 4. On the test set, our model is competitive with GPT-4o, beating or tying its performance on 5 of the 12 classes. On the validation set, we outperform GPT-4o in all but the "water_any" class, showing strong in-sample performance.

Compared to the open source LLaVA-NeXT, GPT-4o broadly outperforms. Since the details of GPT-4o's training and architecture are not public, we can only speculate on the performance gap. We believe that one contributor to the performance gap may be access to sufficient openly available high-quality data for the given domain, thus demonstrating a need for high-quality, open-access labeled disaster-related aerial imagery. Though GPT-4o clearly highlights the potential of multimodal vision-language models for computer vision tasks—particularly in its demonstrated performance across the different distributions of the validation and test sets—its closed-source nature and restrictive licensing makes it difficult to build derivative works.

GPT-4o also suffers from practical challenges for operational use. It is only accessible over the internet via an API, and we found that the API frequently timed out when trying to download large high-resolution images, such as those captured by CAP. This required the images to be downloaded, resized, re-encoded, and uploaded in small batches. As such, it is not suitable for large-scale or time-critical tasks, nor for usage in offline environments. In comparison, our pretrained classifiers run much quicker than both VLMs, and do not require an internet connection like GPT-4o.

We also note that in many classes, we observe performance degradation between the validation and the test set. At first glance, this may be considered evidence of overfitting; however, we contend that most of the difference in performance should be attributable to the distribution shift, since we observe a significant performance difference even in the zero-shot vision-language classifiers, which have not seen the training data. The magnitude in difference in performance of our supervised model is generally comparable to that of zero-shot LLaVA-NeXT from validation to test.

Whereas our baseline model was trained with "standard" supervised image classification techniques, we anticipate that approaches that incorporate domain adaptation techniques, or more recent architectures, such as multimodal language models, should handle the distribution shift better than our model. The baseline model can thus be used in conjunction with the LADI v2 validation and test sets to benchmark the efficacy of such approaches.

| | Test | | | Validation | | |
|---|---|---|---|---|---|---|
| Class | Ours | GPT-4o | LLaVA | Ours | GPT-4o | LLaVA |
| bridges_any | 0.53 | 0.52 | **0.56** | **0.65** | 0.59 | 0.44 |
| buildings_any | **0.94** | **0.94** | 0.88 | **0.96** | 0.93 | 0.94 |
| buildings_affected_or_greater | 0.50 | **0.66** | 0.45 | **0.76** | 0.74 | 0.56 |
| buildings_minor_or_greater | **0.59** | 0.55 | 0.21 | **0.68** | 0.50 | 0.38 |
| debris_any | **0.46** | 0.40 | 0.40 | **0.66** | 0.55 | 0.46 |
| flooding_any | 0.53 | **0.61** | 0.35 | **0.79** | 0.73 | 0.50 |
| flooding_structures | **0.43** | 0.39 | 0.11 | **0.78** | 0.72 | 0.28 |
| roads_any | 0.90 | **0.92** | **0.92** | **0.94** | 0.90 | 0.92 |
| roads_damage | 0.17 | **0.18** | 0.07 | **0.44** | 0.18 | 0.18 |
| trees_any | 0.93 | **0.97** | 0.95 | **0.96** | **0.96** | **0.96** |
| trees_damage | 0.45 | **0.50** | 0.26 | **0.75** | 0.54 | 0.55 |
| water_any | 0.79 | **0.84** | 0.82 | 0.91 | **0.93** | 0.89 |

Table 4: Comparison of $F_1$ scores of our method, LLaVA-NeXT, and GPT-4o on the test and validation sets across classes.

## 5    CONCLUSION

*Summary of Contributions.* In this paper, we introduce the LADI v2 dataset, a curated collection of post-disaster aerial images from multiple perspectives, hazard types, and geographies across the United States. The dataset addresses the need for high-quality, diverse training data to support the development of computer vision models for disaster response applications. To facilitate research and implementation efforts, we provide the dataset and two pretrained reference classifiers as open-source resources.

One of the key strengths of LADI v2 is the quality of its annotations, which were provided by trained Civil Air Patrol volunteers using label sets and training materials developed in collaboration with FEMA. This ensures that the labels are consistent with the standards used by emergency management professionals, enhancing the practical utility of the dataset.

Furthermore, LADI v2 features a realistic distribution shift between the training, validation, and test splits, capturing the year-to-year variability in disaster events as well as changes in operational procedures and technology, such as the increased adoption of the WaldoAir system by the Civil Air Patrol. This characteristic makes the dataset a valuable benchmark for evaluating domain adaptation techniques in the context of disaster response.

The pretrained classifiers demonstrate strong performance on the LADI v2 test set, outperform state of the art open source open-vocabulary classification from LLaVA-NeXT, and are competitive with commercial offerings such as GPT-4o on the most relevant damage classes. This comparison highlights the value of open source domain-specific training data for specialized applications and underscores the potential impact of the LADI v2 dataset on the broader machine learning community.

*Limitations and potential improvements.* While LADI v2 represents a step forward in the availability of high-quality annotated disaster imagery datasets, there are some limitations to consider. First, certain hazard types and geographies are under-represented due to multiple factors, including the likelihood of those hazard types occurring, the likelihood that a federal disaster declaration is issued, and whether CAP is tasked to collect images. For example, imagery from California is comparatively underrepresented in the dataset relative to the state's size, population, and exposure to hazards because its state-level emergency management agency is relatively well-equipped, reducing the likelihood of FEMA-supported CAP missions in the area.

Second, LADI v2 is specific to the United States, which means that the architecture, biomes, disaster types, and infrastructure represented in the dataset is not representative of the rest of the world. Researchers and practitioners working on disaster response applications outside of the US should augment the dataset with additional imagery from the application domain.

Finally, LADI v2 currently supports only multi-label classification tasks. Applications requiring finer-grained localization or segmentation may require additional effort to adapt the dataset or models to their specific needs. The alignment of building damage labels to existing datasets such as those in Manzini et al. (2024); Rahnemoonfar et al. (2022); Gupta et al. (2019) offers a possibility of combining LADI v2 with those other datasets in multi-task training frameworks.

Despite these limitations, LADI v2 offers a valuable resource for the machine learning community and has the potential to support the development of more effective and efficient disaster response tools. Future work could expand the dataset to include a wider range of geographies and disaster types, as well as provide additional annotation classes to support other vision tasks.

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

# 6 APPENDIX

## 6.1 V1 AND V2 COMPARISON CLASS MAPPINGS

| Original Class | Source Dataset | Mapped Class(s) |
|---|---|---|
| flood | v1 | flooding, damage |
| rubble | v1 | damage, debris |
| misc_damage | v1 | damage |
| building | v1 | building |
| road | v1 | road |
| bridges_any | v2 | |
| buildings_any | v2 | building |
| buildings_affected_or_greater | v2 | building, damage |
| buildings_minor_or_greater | v2 | building, damage |
| debris_any | v2 | damage, debris |
| flooding_any | v2 | flooding, damage |
| flooding_structures | v2 | building, flooding, damage |
| roads_any | v2 | road |
| roads_damage | v2 | damage |
| trees_any | v2 | |
| trees_damage | v2 | damage |
| water_any | v2 | |

Table 5: The class mappings established between the LADI v1 and v2 labels and the condensed label set for Section 4.2

## 6.2 SEMANTIC SIMILARITY ANALYSIS

CLIP (Contrastive Language-Image Pretraining) (Radford et al., 2021) is a vision language model pretrained on 400 million image-text pairs from the internet. The model is given a batch of images and captions, and is trained to pair the associated image to its respective caption. In doing so, the model learns to align the encoded image representation to the respective encoded caption text representation. As a result, images with similar textual descriptions tend to be closer together in the CLIP image embedding space, and dissimilar images are further apart. We use this property to characterize the distribution of our validation and test sets below.

We attempt to quantify "out-of-sample-ness" by using distance in CLIP space (Radford et al., 2021). CLIP embeddings align images with similar textual descriptions, such that images with semantically similar content will be nearby in CLIP space even if they are not necessarily visually similar in their pixel representations. In this way, we can use distance in CLIP space as a proxy for semantic similarity between two images, where similar images are closer in CLIP space. We use the Euclidean distance between normalized vectors as the distance metric, $d(\theta) = \sqrt{2(1 - \cos\theta)}$, where $\theta$ is the angle between the two vectors. For each image in the validation and test sets, we compute the CLIP distance between it and its nearest neighbor in the training set. We also compute the $L^1$ norm of the error vectors (the difference between the post-sigmoid/pre-threshold prediction and ground-truth vectors) for each image in the validation and test sets.

We visualize the joint distribution of the $L^1$ error norm and distance to nearest training point for each image in the validation (blue) and test (orange) set in Figure 5. Kernel density estimates of the marginal distributions are visualized along the top and right hand axes. We can see that the test set is on average further away in CLIP space and has larger error norms. There appears to be a positive relationship between the distance to the nearest training example and the average error norm, as well as the variance in the distribution of the error norm. This approach could be used characterize how out-of-sample a given set of images is, as well as estimate the potential expected degradation of performance associated with that distribution shift.

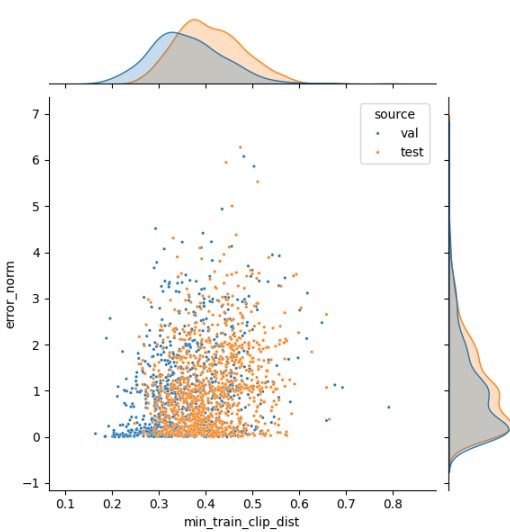

Figure 5: Error vector $L^1$ norm vs. the distance from a point in an evaluation set to its nearest neighbor in the train set in CLIP space. Validation data is plotted in blue and test data in orange.

