# OpenReview forum: "LADI v2: Multi-label Dataset and Classifiers for Low-Altitude Disaster Imagery"
_ICLR.cc/2025/Conference — ICLR 2025 Conference Withdrawn Submission_

### Official Review · Reviewer_cgxP · 2024-10-18

**Soundness:** 3
**Presentation:** 3
**Contribution:** 2
**Rating:** 3
**Confidence:** 3

**Summary:**

This work introduces a dataset of post-disaster aerial images with multi-label annotations (12 categories, e.g. flooding, damaged bridges, destroyed buildings). The images vary in viewing angle, geographical location, and disaster type. The intended use of this data is to train computer vision models to rapidly filter post-disaster imagery for actionable information. The paper also evaluates two transformer models as baselines for the multi-label classification task. Two publicly available vision-language models (LLaVa, GPT-4o) are also evaluated.

Overall, this seems like a nice dataset for an important problem. With some additional work, I think it could be a solid contribution. In its current form, the technical contributions are modest.

**Strengths:**

* Important problem. Making post-disaster imagery rapidly searchable sounds like an excellent use case for computer vision.
* Good labeling protocol. The labels are provided by an end-user (FEMA / Civil Air Patrol) who provides their annotators with standardized training. Multiple labelers review each image.
* The temporal split fairly models the real use-case: training a model with data up to time $T$ and applying it for $t > T$.
* Comparison between supervised models and vision-language (GPT-4o and LLaVA) models are interesting (vision-language models like GPT-4o achieve reasonable but generally worse performance).
* Clear hyperparameter tuning procedures.
* Related work seems to be covered well, though I am not deeply familiar with the area and will defer to more knowledgeable reviewers on this point.

**Weaknesses:**

The machine learning content of this work is modest. While there are some interesting experiments (e.g. the vision-language model comparisons), the benchmarking consists of only two supervised models and two off-the-shelf VLMs. Further, mAP values are not reported for the VLMs so one cannot compare them to the supervised models directly on the primary benchmark metric. Even then, only one of the two supervised models is evaluated in the latter half of the paper.

The authors mention that 25 models were trained, but those results are not provided. This is unfortunate, since a comprehensive analysis of those models would significantly strengthen the paper if it provided technical insights into distribution shift, hyperparameter sensitivity, VLM strengths/weaknesses, etc.

Section 4.2 claims to "assess the marginal benefit of using trained annotators over crowdsourced workers" - however, there seem to be many confounding differences between the two datasets (not least of which is that the two datasets consist of almost entirely different images). How could it be possible to attribute differences in model performance to the label quality?

**Questions:**

1. What are the mAP scores for the VLMs?
2. Why are results for the 25 trained models mentioned in the paper not included?
3. What are the properties of this dataset that are not covered by other related datasets?
4. Why was the validation set IID instead of temporal? Why not use a temporal split like for the test set?
5. What are the use cases for multi-label compared to finer annotations like segmentation? Is one a definitively better cost-benefit proposition for the end users?
6. Can you specify the result in Beyer (2022) that supports the pretraining observations (L338). It wasn't obvious on a quick skim, but I may have missed it.

**Details Of Ethics Concerns:**

High-stakes application area. Mistakes made by the model can have major consequences for life and property.

---

> ### Author Response · Authors · 2024-11-12
> **Answer to questions**
>
> 1- The VLMs don't provide a confidence score, so we cannot compute mAP
>
> 2- These can be included in supplementary information
>
> 3- Expert annotation, oblique post-disaster imagery
>
> 4- We chose validation to be IID to training so that they could be directly compared to ensure we're not overfitting
>
> 5- The use case that motivated the dataset is based on filtering images for analyst review. The volunteers had previously been trained in this filtering task, but did not have training on finer-grained annotations, such as detection or segmentation. The end-users also did not express a need for finer-grained annotations, so it was determined to be outside of our scope for this project.
>
> 6- From Beyer 2022, section 4.1:
> "Not only is there limited benefit of training a large model size on a small
> dataset, but there is also limited (or even negative) benefit from training a small
> model on a larger dataset. Perhaps surprisingly, the ResNet-50x1 model trained
> on the JFT-300M dataset can even performs worse than the same architecture
> trained on the smaller ImageNet-21k. Thus, if one uses only a ResNet50x1,
> one may conclude that scaling up the dataset does not bring any additional
> benefits. "

---

### Official Review · Reviewer_QiAd · 2024-10-28

**Soundness:** 2
**Presentation:** 2
**Contribution:** 2
**Rating:** 3
**Confidence:** 5

**Summary:**

The authors propose a low-altitude dataset of ~10,000 post-disaster imagery.  They also pretrain two baseline classifiers.

**Strengths:**

Rapid response to disasters is an important area where ML can contribute, so having a sound dataset for model development and training is valuable to the community.

**Weaknesses:**

The exact goal that the dataset is enabling is a bit unclear.  The authors state that when a mission is flown, there are a lot of images, but not all of them have relevant (i.e. damage) in them.  However, what that enables (and therefore what type of data is needed and what performance is required) is not clear- are they trying to find a specific building to visit on the ground (in that case isn't more precision needed), are they trying to estimate the magnitude of the damage (in that case doesn't the imagery need to be de-dupped)?

The authors mention that these annotators were more experienced than those for LADIv1, but additional information around training and (dis)agreement between annotators is necessary to speak to the quality of the annotations.

Some kind of analysis showing the performance out of distribution is important given that this data is limited to certain areas.  Showing the generalization capabilities is important for establishing how it could be used in a new setting.

GPT-4o, with no knowledge of this dataset, does a reasonably good job, if not better.  Therefore the contribution from the models is somewhat limited.

**Questions:**

Given the performance of GPT-4o, could LADI v1 be improved (and combined with LADI v2) for improved results?

---

### Official Review · Reviewer_P14g · 2024-10-30

**Soundness:** 2
**Presentation:** 3
**Contribution:** 1
**Rating:** 3
**Confidence:** 4

**Summary:**

The paper proposes a new dataset with around 10k images of disaster events with multi-label classes. The paper improves on a past dataset by collecting labels from experts in the area called civil air patrollers. The dataset is divided into a training and testing set based on the year of acquisition. Two different sets of models are tested on the dataset one that is fine-tuned to perform classification as well as zero-shot multimodal LLMS.

**Strengths:**

The dataset has the potential to be very relevant for the use of disaster reporting. It covers diverse types of events, object types, time period, and locations.

**Weaknesses:**

Not a challenging benchmark:
   * The numbers in the results suggest that the benchmark is not very challenging. Most simple fine-tuning methods can reach up to 90% mAP on the test set. Even zero-shot methods can perform well. This suggests that while the benchmark might be useful for the disaster reporting community, it is not very useful for ICLR, since all the proposed methods have already saturated the performance. Some ways to make the dataset more challenging could be adding extreme disaster events, or adding other tasks such as few-shot learning.
   * The dataset only contains classification labels which are fairly well explored in the learning community. Other types of labels such as detection, segmentation, or captioning would have created a more challenging benchmark for the community. However, the good performance of zero-shot and fine-tuned models suggests the benchmark is not very challenging.

Potential improvements in benchmarking:
   * The number of examples in the test/training set while around 1k/10k is not huge. Therefore, it might be better to train models multiple times and provide an error bound around the numbers. With 1k images in the test set, there is a possibility that the errors are huge.
   * It is not clear how much time and effort it takes for CAPs to label the entire dataset. When creating a new dataset this cost should be mentioned, so that future researchers have an estimate of cost to them in case of collecting more data.
   * A difference between LADI v1 and v2 is that v1 was collected by a layperson while v2 was collected by experts. Expert labels might matter for domains where images are not well represented on the internet. The experiments with GPT and LLAVA suggest that these images are well-represented. Secondly, costwise collecting expert labels is typically more expensive than a layperson. Therefore, it might be useful to do the same cost analysis, i.e. with a larger set of data collected with layperson labels, to truly show the effectiveness of this method.
    * More qualitative examples should be shown. For example, Figure 1 can be labeled with the true labels. Qualitative examples can be shown where certain models fail and others do not.
    * Some scaling law analysis is also needed: How many images are needed to be collected to get to a good performance? is it less than 10k or a lot more?

Many arguments in the paper are presented but not confidently resolved:
   * For example, the reason for the distribution shift (line 275) is attributed to both a change in technology (waldoair) as well and changes in event types. While such distribution shifts should be present in challenging benchmarks, the shifts should be studied in more detail. For example, one way to study this would could be to create a test set from events of some other year, when Waldoair was not introduced.

   * Line 456, suggest a significant performance difference between test and val set for ZS methods. However, it is not directly evident from the table. To me it seems like ZS methods indeed generalize better to the test set compared to the fine-tuning methods.

**Questions:**

In the co-occurrence matrix, it is hard to understand what these numbers are a percentage of. neither the rows, not the column, nor the whole matrix sums up to 100. How the normalization is done and what these numbers mean should be explained in the paper.

Minor suggestions:
In Figure 2a some of the text is covered by Figure 2b. This should be fixed.
The maps in general are hard to see/understand/sometimes misleading:
   * In Fig 2b the states not considered can be grayed out like 4b. This makes it seem like all states have some images which might not be true.
   * Hawaii can be cropped and zoomed in on more.
   * Puerto Rico can also be placed in an inset and zoomed in more.
   * The figures need improvement. Many of the fonts are hard to read/understand.


The quotations look weird in many places such as line 100: in latex use '' and `` for left and right quotes.

---

### Official Review · Reviewer_V2B5 · 2024-11-03

**Soundness:** 3
**Presentation:** 2
**Contribution:** 3
**Rating:** 6
**Confidence:** 4

**Summary:**

This paper introduces a new multi-label image classification benchmark for natural disasters. During the benchmark construction, the authors extensively involved experts to ensure the practicality of the dataset. Extensive analyses are conducted to show the usefulness of the dataset.

**Strengths:**

[S1] Expert involvement during development: The construction of the dataset involves heavy expert involvement (from annotating images to deciding classes), ensuring the practicality of the benchmark.
[S2] Focus on low-altitude imagery: As described in the related work, prior datasets on low-altitude imagery are limiting
[S3] Extensive analysis: The domain gap between val and test is large (table 4) enough that it could be an interesting machine learning challenge to figure out how to bridge the domain gap.

**Weaknesses:**

[W1] Open-vocabulary classification: The authors make a comparison to several SOTA generalist VLMs. However, since this is a highly specialized domain, it would be useful to compare to VLMs that are developed for remote sensing [1, 2]


[1] Mall, Utkarsh, Cheng Perng Phoo, Meilin Kelsey Liu, Carl Vondrick, Bharath Hariharan, and Kavita Bala. "Remote sensing vision-language foundation models without annotations via ground remote alignment." arXiv preprint arXiv:2312.06960 (2023).

[2] Liu, Fan, Delong Chen, Zhangqingyun Guan, Xiaocong Zhou, Jiale Zhu, Qiaolin Ye, Liyong Fu, and Jun Zhou. "Remoteclip: A vision language foundation model for remote sensing." IEEE Transactions on Geoscience and Remote Sensing (2024).

**Questions:**

Questions:
[Q1] What is the GSD of the imagery? If the GSD is small, has care been taken to ensure the residents' privacy?

**Details Of Ethics Concerns:**

The benchmark might involve imagery that could contain sensitive private information since they are captured by UAVs. There is no mention of the ground sample distance of the imagery and any special measures to protect residents' privacy. The reviewer has asked the authors to clarify the issue. Depending on their response, further review might be needed.

---

### Note · Authors · 2024-11-12

I have read and agree with the venue's withdrawal policy on behalf of myself and my co-authors.